# Highly Hydrophilic TiO_2_ Nanotubes Network by Alkaline Hydrothermal Method for Photocatalysis Degradation of Methyl Orange

**DOI:** 10.3390/nano9040526

**Published:** 2019-04-03

**Authors:** Jin Yang, Jun Du, Xiuyun Li, Yilin Liu, Chang Jiang, Wenqian Qi, Kai Zhang, Cheng Gong, Rui Li, Mei Luo, Hailong Peng

**Affiliations:** 1Key Lab of Poyang Lake Environment and Resource Utilization (Ministry of Education), Nanchang University, Nanchang 330031, China; yangjin199501@sina.com (J.Y.); show7898@foxmail.com (X.L.); 15079171252m0@sina.cn (Y.L.); 16607001729y@sina.com (C.J.); qiwenqian9873@sina.com (W.Q.); zhangkai7755@sina.com (K.Z.); stokescheng@sina.com (C.G.); lr3132@sina.com (R.L.); luomei@ncu.edu.cn (M.L.); 2Jiangxi Province Key Laboratory of Edible and Medicinal Plant Resources, Nanchang University, Nanchang 330031, China; 3Department of Chemical Engineering, School of Environmental and Chemical Engineering, Nanchang University, Nanchang 330031, China

**Keywords:** titanium dioxide, nanotubes network, photocatalysis, alkaline hydrothermal method, hydrophilicity

## Abstract

High-density and highly cross-coated anatase TiO_2_ nanotubes networks have been successfully prepared on the surface of Ti foil by alkaline hydrothermal using NaOH and Ti foil as the precursors. The nanotubes networks were analyzed using X-ray diffraction (XRD), energy dispersive X-ray spectrometer (EDX), transmission electron microscope (TEM), scanning electron microscopy (SEM), optical contact angle tester, and ultraviolet (UV) fluorescence spectrophotometer, respectively. The results showed that the nanotubes network with diameters of 30–50 nm were obtained on the Ti foil surface. The morphology of the nanotubes network possessed the three-dimensional network structure, The TiO_2_ nanotubes network grew along the (101) direction of the tetragonal anatase crystal. The morphology and crystal phase of the TiO_2_ nanotubes network were better at the conditions of NaOH concentration 7–10 mol/L and temperature 160–170 °C. The best contact angle of TiO_2_ nanotubes network after UV-light irradition was only 5.1 ± 2.9°. Under the irradiation of mercury lamp, the nanotubes network exhibited excellent photocatalytic performance and the degradation ratio of methyl orange solution reached to 80.00 ± 2.33%. Thus, the anatase TiO_2_ nanotubes network has great potential in applications for pollution photocatalytic degradation.

## 1. Introduction

In recent years, a large number of pollutants have been found in various types of water bodies [1], from surface water to groundwater resources [2,3]. Therefore, how to deal with water pollution has become an urgent problem to solve. Photocatalysis is a new technology developing rapidly in recent years to make use of solar energy for environmental purification and energy conversion [4]. As a photocatalyst, TiO_2_ can completely and rapidly degrade various toxic compounds and oxidize acids, dyes, organic phosphorus pesticides, fuel oil, and other organic N-containing compounds in water into CO_2_, H_2_O, and other non-toxic substances [5]. Therefore, TiO_2_ has been widely used in the fields of water treatment and air purification for its features of chemical stability, insolubility, light corrosion resistance, non-toxicity, and low cost [6,7,8,9,10,11,12,13].

Nowadays, different forms of nano-TiO_2_ have attracted extensive attention due to their improved toxic compounds degradation and photocatalytic performance, such as nanoparticles [14], nanobelts [15], nanowires [16], nanorods [17], nanotubes [18] and nanoflowers [19]. Meanwhile, nanotubes not only have chemical stability but also superior photocatalytic properties. Therefore, more and more researchers are doing research into nanotubes. Sreekantan et al. used commercial titania nanoparticles as starting material to carry out hydrothermal reaction in aqueous sodium hydroxide solution, and the prepared anatase TiO_2_ nanotubes exhibited excellent photocatalytic performance [20]. Sun et al. successfully prepared TiO_2_ nanotubes on carbon fibers substrate using TiO_2_ nanowires as reaction materials [21]. Up to now, various methods have been used to prepare nanotubes, such as templating synthesis method [22], sol-gel method [23], anodization method [24], and hydrothermal method [25]. Among them, the hydrothermal method has the advantages of low reaction temperature and easy to obtain. Additionally, the nanotubes prepared by the hydrothermal method have high purity and a good crystal phase. Therefore, hydrothermal synthesis has a broad developing prospect in preparation of nano-materials [26].

At present, the main preparation substrate of TiO_2_ nano-materials were the glass, absorbent (activated carbon, Silica gel, etc.) and metal. However, glass when used as a substrate has its limitations of poor combination degree with TiO_2_, leading to TiO_2_ being easily detached from the substrate. The absorbents were used as a substrate, which not have conductivity. In order to overcome the disadvantages of these substrate, Ti foil has been used as the TiO_2_ nano-materials precursor and the substrate. Ti foil has excellent conductivity and is not easy to break. TiO_2_ nano-materials were directly grown from Ti foil, not only made it avoid the complicated preparation of precursor reactants, but also can fix tightly on the Ti foil. Moreover, the TiO_2_ nano-materials grown from Ti foil have many advantages of high yield and high density. Thus, Ti foil was the excellent substrate for the practical application of TiO_2_ nanomaterials.

In this study, high-density anatase TiO_2_ nanotubes network were successfully prepared using alkaline hydrothermal method on the surface of Ti foil. The novelty of this work has been demonstrated as follows: (1) the TiO_2_ nanotubes network are grown directly on the Ti foil surface using alkaline hydrothermal method with excellent photocatalytic and hydrophilic properties; (2) the TiO_2_ nanotubes network and Ti foil are tightly connected, which facilitates recycling in the application to prevent secondary pollution; (3) the effect of NaOH concentration and hydrothermal temperature on TiO_2_ nanotubes network morphology and properties has been demonstrated.

## 2. Experiment

### 2.1. Materials

Ti foil was purchased from Tenghui Titanium Factory (Jinhua, China). Sodium hydroxide (NaOH) and Anhydrous ethanol (C_2_H_5_OH) were obtained from Tianjin Da Mao chemical reagent factory (Tianjin, China). Methyl orange (MO) was purchased from Shanghai Zhanyun Chemical Co., Ltd. (Shanghai, China). Acetone (C_3_H_6_O) was purchased from Shanghai Yanchen Chemical Industrial Co., Ltd. (Shanghai, China). The deionized water was obtained from a Mili-Q Ultrapure water system (Millipore, Bedford, MA, USA).

### 2.2. Preparation of TiO_2_ Nanotubes Network

The Ti foil was successively placed in acetone solution, anhydrous ethanol and deionized water for ultrasonically cleaned to remove the oxide and residual oil on the surface of the Ti foil. NaOH (60 mL, 1–10 mol/L) and Ti foil (50 × 30 × 3 mm) were added to the polytetrafluoroethylene liner, respectively. It was noted that the Ti foil must be lie vertically on the edge of the polytetrafluoroethylene liner. It can increase the contact area between Ti foil and NaOH. Next, the polytetrafluoroethylene liner was placed in a high pressure reactor for 4 h in the reaction chamber at a temperature of 130–170 °C. When the high pressure reactor reached room temperature, the Ti foil was taked out from the polytetrafluoroethylene liner, and then rapidly added into HCl solution (0.1 mol/L) for pickling for 12 h. Finally, the Ti foil was dried in a vacuum drying oven at 50 °C and was inserted into the muffle furnace (500 °C) to anneal for 2 h. After that, the TiO_2_ nanotubes network was formed. The preparation process was as follows (Scheme 1).

### 2.3. Characterization

X-ray diffractometry (XRD, Bede, Durham, UK) was performed to characterize the phase structure of the samples, X-ray diffractometer with CuKα (λ = 0.15406 nm) radiation operated at 40 kV and 35 mA, Diffractograms were collected in between 10° to 8°. A field emission scanning electron microscope (FESEM, FEI Quanta200F, FEI, Hillsboro, OR, USA) was used to identify the morphology of the as-deposited material, experimental testing requires high-vacuum conditions, the sample was cut into a square with a side length of 10 mm, the sample is closely attached to the sample stage with conductive tape, and the sample is amplified by different multiples for detection. High-resolution transmission electron microscope (HRTEM) images were obtained on a JEOL-2010 HRTEM (JEOL, Tokyo, Japan) using an acceleration voltage of 200 kV, TEM grids were prepared by dispersing samples in Anhydrous ethanol (5 min ultrasonication) and then dropping onto a carbon Cu grid. The sizes of nanotubes network were determined by the scale plate in the TEM and SEM images. The photocatalytic reaction was carried out in a photochemistry reaction instrument (BL-GHX-V, BILON, Shanghai, China), photochemistry reaction instrument (480 mm × 420 mm × 900 mm) is a sealed rectangular parallelepiped structure, The sample was added into a quartz reaction tube with 35 mL MO (20 mg/L) for photocatalytic degradation, and a 500 W high pressure mercury lamp was used as the light. The hydrophilic properties were observed on the optical contact angle measuring instrument (Zheke, DSA 100, Zheqi Technology, Beijing, China), Dropping a drop of deionized water on the sample surface was employed by the hanging drop method to determine the contact angle of the droplet with the sample.

### 2.4. Photocatalytic Reactions of TiO_2_ Nanotubes Network

The TiO_2_ nanotubes network sample was cut into a square with a side length of 10 mm, which was added into a quartz reaction tube with 35 mL MO (20 mg/L) for photocatalytic degradation. MO (20 mg/L) standard solution with the concentration of 1–40 mg/L had a good linear relationship with absorbance. The 500 W high-pressure mercury lamp with a main wavelength of 365 nm was used as the light source. The effective area of TiO_2_ nanotube network for photocatalytic reaction was 1 cm^2^. The light intensity was measured as 160 mW/cm^2^ in the reaction system. This was taken out the MO solution every 15 min and tested for absorbance. The photocatalytic performance is determined by the degradation rate (*η*), which is calculated as follows [27]:(1)$$\eta =\frac{C_{0}-C}{C_{0}}\times 100\%=\frac{A_{0}-A}{A_{0}}\times 100\%$$where *C*_0_ and *C* are the initial concentration and the concentration of MO at different time, *A*_0_ and *A* are the initial absorbance and absorbance of MO at 466 nm under ultraviolet (UV) light irradiation at different time.

## 3. Results and Discussion

### 3.1. Formation and Photocatalytic Mechanism of TiO_2_ Nanotubes Network

The formation mechanism of TiO_2_ nanotube network can be explained by the following three stages. Firstly, Ti foil can react with NaOH to form titanate hydrogel, the fracture of titanate hydrogel is recombined into layered Na_2_Ti_3_O_7_. As the reaction proceeded, the layered Na_2_Ti_3_O_7_ continuous grown on the Ti foil surface, a multilayer nanotube film covering the Ti foil surface was formed. Then, Na^+^ was replaced by H^+^ in the process of pickling with HCl, and the reaction produced H_2_Ti_3_O_7_. Finally, the H_2_Ti_3_O_7_ was annealed at high temperature, and the anatase TiO_2_ nanotubes network were prepared after dehydration of titanite and lattice rearrangement. The reaction scheme of the TiO_2_ nanotubes network formations is given below:3Ti + 2NaOH + 5H_2_O → Na_2_Ti_3_O_7_ + 6H_2_ (g)(2)
Na_2_Ti_3_O_7_ + 2HCl → H_2_Ti_3_O_7_ + 2NaCl(3)
H_2_Ti_3_O_7_ → 3TiO_2_ + H_2_O(4)

When ultraviolet light irradiated on the surface of TiO_2_, the electron was excited from the valance band to the conduction band. Therefore, highly active photogenerated electrons (e^−^) and holes (h^+^) were produced on the conduction and valence bands. Dissolved oxygen (O_2_) captures the electron to form atomic oxygen (·O^2−^), and the holes oxidize OH^−^ and H_2_O adsorbed on the surface of the catalyst to form hydroxyl radicals (·OH), which can degrade most organic pollutants into carbon dioxide and water, and degrade inorganic pollutants into harmless substances. The photocatalytic mechanisms of TiO_2_ towards the degradation of MO is that the h+ react with H_2_O to form ·OH, ·OH oxidize and degrade MO, and the azo bond and benzene ring of MO are broken to produce H_2_O, CO_2_, SO_4_^2^^−^ and NO_3_^−^ to achieve the degradation effect. The photocatalytic mechanism of TiO_2_ is given below:TiO_2_ + hv → TiO_2_ (h^+^ + e^−^)(5)
h^+^ + OH^−^ →·OH(6)
h^+^ + H_2_O →·OH + H^+^(7)
e^−^ + O_2_ →·O^2−^(8)

### 3.2. Characterizations of TiO_2_ Nanotubes Network

Figure 1 shows the XRD patterns of different reaction stages of hydrothermal reaction. According to the XRD standard card of Ti (PDF NO. 44-1294), the diffraction peaks located at 38.42°, 40.17°, 53.0°, 62.94° and 70.66° was correspond to the (002), (101), (102), (110), and (103) planes of Ti (Figure 1a). The diffraction peaks of samples located at 24.32°, 28.35° and 47.78° without washing with HCl, which correspond to the (102), (111) and (020) planes of Na_2_Ti_3_O_7_ (Figure 1b), according to the XRD standard card of Na_2_Ti_3_O_7_ (PDF NO.72-0148). After the hydrothermal reaction, the diffraction peaks located at 24.37° and 48.52°, which correspond to the (102) and (020) planes of H_2_Ti_3_O_7_ (Figure 1c). After the sample was washed with HCl and high-temperature annealing, the diffraction peak of titanate was disappeared and the diffraction angles (2θ) of 25.3°, 48.04°, 53.88° and 55.06° correspond to the (101), (200), (105) and (211) planes for anatase TiO_2_ (Figure 1d). These results showed that an anatase TiO_2_ crystal phase has formed.

Scanning electron microscope (SEM) micrographs of the samples prepared at different reaction stages of hydrothermal reaction are shown in Figure 2. Figure 2a showed that the surface of the Ti is not flat with some steps and cracks, but there is no a linear structure. From Figure 2b, it can be seen that the Ti foil surface was dense and irregular Na_2_Ti_3_O_7_ with the size of about 2 μm. Figure 2c showed that H_2_Ti_3_O_7_ was obtained after washing with HCl, the Ti foil surface was covered with dense and uniform nanotubes, and the growth of the nanotubes has no fixed direction. The nanotubes were randomly grown on the surface of the Ti foil to form a network structure, and the diameter of the nanotubes were approximately 30 nm. Figure 2d showed the micrograph of the TiO_2_ after the sample was high-temperature annealed. Compared with the network structure of H_2_Ti_3_O_7_ nanotubes, the network structure of TiO_2_ nanotubes were more slender and looser, and have better cross-over in the longitudinal direction. The network structure of TiO_2_ nanotubes can help internal TiO_2_ nanoparticles to more use during the reaction processes.

Figure 3 showed the XRD patterns of the TiO_2_ nanotubes network prepared on Ti foil surface with different NaOH concentrations at 160 °C for 4 h. Obviously, four strong diffraction peaks at 2θ values of 38.42°, 40.17°, 53.0° and 62.94° were found in Figure 3. According to the XRD standard card of pure Ti (PDF NO. 44-1294), the diffraction angles (2θ) of 38.42°, 40.17°, 53.0° and 62.94° correspond to the (002), (101), (102), and (110) planes for Ti, respectively. This indicated that the Ti substrate always exists. Additionally, new diffraction peaks appeared at 2θ of 25.3° and 48.04° and its intensity increased with the concentration of NaOH solution increasing. Compared with the XRD standard card of TiO_2_ (PDF NO. 78-2486), it is obvious that the diffraction angles (2θ) 25.3° and 48.04° were separately characterized as the (101) and (200) planes for anatase TiO_2_, respectively. This result showed that the anatase TiO_2_ crystal phase was formed. The Ti foil was not enough to react with the low concentration of NaOH to form titanate hydrogels until the concentration of NaOH solution was increased to 3 mol/L. When the NaOH concentration increases from 3 mol/L to 10 mol/L, the content of TiO_2_ crystal continuously increased. These results indicated that a certain thickness of anatase TiO_2_ crystal was formed in the Ti foil surface.

SEM micrographs of TiO_2_ nanotubes network were shown in Figure 4 with different NaOH concentrations. As shown in Figure 4, the flake or layered structure was gradually transformed into more elongated nanotubes, and the nanotubes were interwoven together to form a network structure. Low NaOH concentration is not sufficient to allow the Ti foil dissolved continuously. When the NaOH concentration increased to 5 mol/L, the layered film was cracked into a bundle structure interwoven together, and each bundle had a diameter of about 100 nm (Figure 4c). When the NaOH concentration continue to increase, the nanotubes with diameter of 50 nm can be observed (Figure 4d). At this time, the nanotubes were structurally intact and uniform size, and the nanotubes interwoven together to form micropores with the diameter of 100 nm. The density of the nanotubes became larger when the NaOH concentration increased to 10 mol/L, and the nanotubes became more slender with diameter of about 30 nm (Figure 4e). Obviously, more Ti react as the concentration of NaOH increases. Meanwhile, the density of TiO_2_ nanotubes gradually increased with the NaOH concentration increasing, which was consistent with the result of XRD.

Figure 5 shows the XRD patterns of TiO_2_ nanotubes network prepared on Ti foil surface with different hydrothermal temperature under conditions of 7 mol/L NaOH for 4 h. According to the XRD standard card of TiO_2_ (PDF NO. 78-2486), the diffraction peaks located at 25.3° and 48.04° corresponded to the (101) and (200) planes for anatase TiO_2_ (Figure 5). When the hydrothermal temperature increased from 130 °C to 170 °C, the diffraction peaks of anatase TiO_2_ was increased and the diffraction peaks of Ti decreased, which indicated that the TiO_2_ grows with consuming of Ti. When the temperature was 130 °C, the dissolved precursor concentration was low, and it was not good for the anatase TiO_2_ nanotubes network to grow. The Ti dissolution in NaOH solution was accelerated with the increasing of hydrothermal temperature, which makes the reaction faster. Therefore, the temperature increasing can promote growth of the TiO_2_ nanotubes network.

The corresponding morphologies of the TiO_2_ nanotube network prepared at different reaction temperature were shown in Figure 6. When the temperature was 130 °C, the nanotube interwoven together to form deep holes with diameter of about 2 μm. At such a temperature, the Ti foil surface could not be completely reacted. When the temperature continued to increase, the deep holes were getting smaller and smaller, and the deep holes were about 1 μm in diameter at 150 °C. It can be clearly seen that deep holes were slowly filled with TiO_2_ nanotubes (Figure 6b,c). When the reaction temperature increases to 160 °C, the TiO_2_ nanotubes formed, and the diameters of the nanotubes were approximately 50 nm. Meanwhile, the nanotubes intersected to form a highly dense network structure (Figure 6d). When the reaction temperature increases to 170 °C, the deep holes disappeared completely (Figure 6e) and the diameter of the nanotubes little changed, and the nanotubes network on the Ti foil surface was stodgier.

The characterization of the TiO_2_ nanotubes are shown in Figure 7. As shown in Figure 7a, it can be concluded that the nanotubes were composed of Ti and O elements. Cu element and C atom generated from copper grid and carbon film on the copper grid [21], respectively. Some approximately 2 μm long nanotubes with a diameter of 50 nm are shown in Figure 7b, which was in good agreement with the SEM micrographs. The four diffraction rings of d_1_, d_2_, d_3,_ and d_4_ were consistent with the (101), (200), (105) and (211) crystallographic planes of TiO_2_ (Figure 7c), respectively, which was consistent with the XRD standard card of TiO_2_ (PDF NO. 78-2486). Regular lattice fringes can be seen clearly in the high-resolution transmission electron microscope (HRTEM) image and *d*-spacings of crystallographic planes were 0.354 nm, which is consistent with (101) crystallographic planes of TiO_2_ (Figure 7d). According to the XRD pattern and the energy dispersive X-ray (EDX) spectrum, it can be clearly concluded that these nanotubes were anatase TiO_2_ nanotubes.

### 3.3. Hydrophilicity of TiO_2_ Nanotubes Network

Ultraviolet (UV) light stimulates the TiO_2_ surface to produce electron hole pair, which restores Ti^4+^ and oxidizes O^2−^, leading to the generation of oxygen vacancy. The vacancy reacted with surface hydroxyl groups and absorbed H_2_O and form hydroxyl radicals [28]. More and more H_2_O was absorbed by hydroxyl radicals and then presented super-hydrophilic performance [29]. The hydrophilicity of TiO_2_ nanotube network grown at different concentrations of NaOH solution and temperatures is shown in Figure 8. The optical contact angle of the sample was 88 ± 2.9° when the NaOH concentration was 1 mol/L (Figure 8a), and the optical contact angles were 8.9 ± 2.9° and 5.1 ± 2.9° when NaOH concentrations were 7 mol/L and 10 mol/L, respectively. When NaOH concentration increased from 1 mol/L to 7 mol/L, more TiO_2_ crystal phase was produced. Therefore, the hydrophilicity of the TiO_2_ sample increased continuously. The Ti foil surface was completely covered by TiO_2_ crystal phase at NaOH concentration of 7 mol/L, and the physical adsorption and chemical adsorption of water molecules had reached saturation. Meanwhile, the optical contact angle of the sample was 8.9 ± 2.9° and 6.5 ± 2.9 °C when the reaction temperature was 160 °C and 170 °C, respectively. Therefore, the hydrophilicity of the sample was best when NaOH concentration and reaction temperature were 10 mol/L and 160 °C, respectively.

### 3.4. Photocatalytic Reaction of TiO_2_ Nanotubes Network

Figure 9 shows the photodegradation behavior of MO solution in the presence of the TiO_2_ nanotubes network grown at different NaOH concentrations and different reaction temperatures (Figure 9a,b). When the NaOH concentration solution was 1 mol/L, few TiO_2_ crystal phase were prepared by hydrothermal reaction and the ratio of photocatalytic degradation was only 11.00 ± 2.33%. The degradation ratio gradually increased with the NaOH concentration increasing. When the concentration of NaOH solution of hydrothermal reaction was 7 mol/L, the relative concentration of MO reaches the minimum with the degradation ratio of 80.00 ± 2.33%. The photocatalytic degradation ratio of TiO_2_ nanotubes was slightly lower than that of 7 mol/L when the NaOH concentration was 10 mol/L, which was due to the TiO_2_ nanotube density causing the micropore size decrease and inhibiting the reactants from reaching the internal active sites. Therefore, the photocatalytic performance of the TiO_2_ nanotube network was best when the concentration of NaOH solution was 7 mol/L.

As shown in Figure 9b, the degradation ratio of MO was only 29.10 ± 2.33% when the reaction temperature was 130 °C, the increase of reaction temperature was beneficial to improve the photocatalytic performance of the TiO_2_ nanotube network. The crystalline phase content of TiO_2_ was low when the reaction temperature was 130 °C, which is not conducive to the MO photocatalytic degradation. As the reaction temperature increases, TiO_2_ nanotubes become longer and denser and were highly crossed on the Ti foli surface to form a network structure, which leads to a high specific surface promoting adsorption capacity. When the temperature was 160 °C, the nanotubes network structure was formed and provided a larger surface area and more reactive sites; the specific surface area of the sample was 130 m^2^/g. The specific surface areas were 94 m^2^/g and 111 m^2^/g when hydrothermal temperature were 140 °C and 150 °C (Figure 10), respectively. Also, the specific surface area of the sample was significantly higher than the specific surface area of P25 was 50 m^2^/g [30], and thus MO can be effectively degraded with the degradation ratio of about 80.00 ± 2.33%. When the temperature increased to 170 °C, the TiO_2_ particles on the Ti foil surface fully reacted and the dense nanotubes network structure blocks the internal space. At that time, the reactants cannot utilize the internal TiO_2_ particles and then the degradation ratio of MO was 68.20 ± 2.33%. Therefore, the TiO_2_ nanotubes network prepared at 160 °C has the best photocatalytic performance. Meanwhile, the TiO_2_ nanotubes network was compared with TiO_2_ photocatalyst (P25). It was found that the photocatalytic degradation ratio of the TiO_2_ nanotubes network for MO was better than that of P25 (Figure 9c).

## 4. Conclusions

In summary, high-density and highly cross-coated anatase TiO_2_ nanotube networks have been successfully prepared by alkaline hydrothermal. The optical contact angle of the TiO_2_ nanotubes network was 5.1 ± 2.9° when the concentration of NaOH was 10 mol/L and the reaction temperature was 160 °C, and the nanotube network reached an optimum with hydrophilic. The crystal phase content, crystallinity, density and morphology of the sample are the key factors affecting the photocatalytic performance, when the concentration of NaOH was 7 mol/L and the reaction temperature was 160 °C. The photocatalytic activity of TiO_2_ nanotubes network reached a maximum, the maximum degradation ratio of MO can reach 80.00 ± 2.33%.

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
