# Peer review of "Highly Hydrophilic TiO2 Nanotubes Network by Alkaline Hydrothermal Method for Photocatalysis Degradation of Methyl Orange"

_nanomaterials, 2019, doi:10.3390/nano9040526_

Reviewer 1 Report

The manuscript can now be published.

Author Response

The manuscript can now be published.

Response: Thank you for your approval.

Reviewer 2 Report

The Authors have adequately addressed all previous points of concern. 

They have added the BET specific surface area data, as requested by Reviewer 2. 

I would only suggest to cite the BET area values with 3 significant digits maximum, i.e., 130, 94 and 111 m2/g (lines 317-320). This would be consistent with the actual uncertainty range of this kind of measurements.

After this very minor point is taken into account, the manuscript could be positively evaluated for publication.

Author Response

Response to Reviewer 2 Comments

We highly appreciate reviewers’ valuable comments, constructive guidance and sincere contributions. We have carefully made specific responses point-by-point and revised the manuscript accordingly in light of the comments, which are annotated in Yellow frame in the revised manuscript.

Reviewers' comments:

The Authors have adequately addressed all previous points of concern. They have added the BET specific surface area data, as requested by Reviewer 2. I would only suggest to cite the BET area values with 3 significant digits maximum, i.e., 130, 94 and 111 m2/g (lines 317-320). This would be consistent with the actual uncertainty range of this kind of measurements. After this very minor point is taken into account, the manuscript could be positively evaluated for publication.

Response: Thank you for your comment. The related content have been corrected into the revised manuscript. (lines 321, 322)

This manuscript is a resubmission of an earlier submission. The following is a list of the peer review reports and author responses from that submission.

Round  1

Reviewer 1 Report

Review Report

Manuscript Number: Nanomaterilas-432780; Title: “Highly Hydrophilic TiO2 Nanotubes Network by Alkaline Hydrothermal Method for Photocatalysis Degradation of MO”. Many parts are well-written and provided scientifically sound methods. Furthermore, there are several points that need to be addressed and revised considerably before acceptance of this paper.

Comments

It would be nice if      authors’ could expand the abbreviation of ‘MO’ in the title of the      manuscript.

Author (s) please      provide BET surface area of TiO2      nanotubes network prepared by

alkaline hydrothermal method.

3.     All SEM images (Fig.3) would be given in the same scale for comparison.

4.     The author(s) should compare the photocatalytic activity of their prepared TiO2 nanotubes with standard photocatalyst in MO degradation for its effciency.

5.     Author(s) would be provided a comparative table showing photocatalysis of TiO2 nanoparticle or nanotube used for the  degradation organic dyes with their current work with other authors’ work with  literature support.

6.     Raman and UV-Vis DRS spectra of prepared TiO2 at different temperature via alkaline hydrothermal method should be presented.

7.     Provide TG analysis of TiO2 Nanotubes. 

8.     Authors’ claimed that TiO2 prepared at 160 oC have the best photocatalytic performance. Please clarify this statement and provide the informations that the surface area and more reactive sites of  TiO2 prepared at this temperature.

9.     Fig.10 should be redrawn and provide the values = Mean ±S.D.

10.  Photocatalytic mechanisms of TiO2 towards the degradation of MO should be described in detail.

Author Response

Response to Reviewer 1 Comments

We highly appreciate reviewers’ valuable comments, constructive guidance and sincere contributions. We have carefully made specific responses point-by-point and revised the manuscript accordingly in light of the comments, which are annotated in RED words in the revised manuscript.

Reviewers' comments:

Title: “Highly Hydrophilic TiO2 Nanotubes Network by Alkaline Hydrothermal Method for Photocatalysis Degradation of MO”. Many parts are well-written and provided scientifically sound methods. Furthermore, there are several points that need to be addressed and revised considerably before acceptance of this paper.

Response: Thank you for your review. I have response your comment point-by-point with red words.

Point 1:

It would be nice if authors’ could expand the abbreviation of ‘MO’ in the title of the manuscript.

Response 1: Thank you for your comment. The imprecise statement was corrected in the revised manuscript (Title).

Point 2:

Author (s) please provide BET surface area of TiO2 nanotubes network prepared by alkaline hydrothermal method.

Response 2: I am so Sorry. BET of TiO2 nanotubes network are temporarily cannot carried out. The Chinese New Year is coming soon and school's testing centre have been closed for holiday. We can provide the experiment and results after Chinese New Yew.

Point 3:

All SEM images (Fig.3) would be given in the same scale for comparison.

Response 3: Thank you for your reminding. The same scale of SEM images were added into the revised manuscript (Figure.3).

Point 4:

The author(s) should compare the photocatalytic activity of their prepared TiO2 nanotubes with standard photocatalyst in MO degradation for its effciency.

Response 4: Thank you for your comments. The related content have been added into the revised manuscript (line 330).

Point 5:

Author(s) would be provided a comparative table showing photocatalysis of TiO2 nanoparticle or nanotube used for the degradation organic dyes with their current work with other authors’ work with literature support.

Response 5: The related content have been added into the revised manuscript (line 295)

Point 6:

Raman and UV-Vis DRS spectra of prepared TiO2 at different temperature via alkaline hydrothermal method should be presented.

Response 6: Thank you for your comments. Raman spectroscopy is mainly used to detect carbon-containing substances. The experimental samples do not contain carbon and therefore, raman is not necessary for our experiment. The forbidden band width of anatase TiO2 is about 3.2 eV, which is basically a fixed value. Therefore, I think it is not necessary to carry out UV analysis.

Point 7:

Provide TG analysis of TiO2 Nanotubes.

Response 7: The TiO2 is a non-combustible material with remarkably high chemical and thermal stability. Therefore, I think it may not be necessary to carry out TG analysis.

Point 8:

Authors’ claimed that TiO2 prepared at 160 °C have the best photocatalytic performance. Please clarify this statement and provide the informations that the surface area and more reactive sites of TiO2 prepared at this temperature.

Response 8: Thank you for your considerable review and comments. The photocatalytic degradation ratio of MO by TiO2 nanotubes were lower than that of 160°C when the temperature were 130-150°C and 170°C. Therefore, TiO2 nanotubes prepared at 160 °C have the best photocatalytic performance (Fiure 7). The number of TiO2 nanotubes were lower than that of 160 °C when the temperature were 130-150°C, meanwhile, more free radicals (active sites) such as h+, ·OH can be produced to photocatalytic degrade of MO.

Point 9:

Fig.10 should be redrawn and provide the values = Mean ±S.D.

Response 9: Thank you for your reminding. The Figure and error bar have been corrected in the revised manuscript (Figure.10)

Point 10:

Photocatalytic mechanisms of TiO2 towards the degradation of MO should be described in detail.

Response 10: The related content have been added into the revised manuscript (line 152).

Reviewer 2 Report

The manuscript describes the synthesis, characterization and one photocatalytic application of a series of TiO2 nanostructures grown over a metal Ti foil.

The text is rather rich in experimental data. However, in its current form, it is not absolutely evident which elements of novelty are present with respect to the previous literature.

Actually, simple and highly reproducible approaches for the preparation of TiO2 nanotubes via hydrothermal method or via anodization of Ti foils have already been extensively described in the last decade (see, for instance, your Ref. 22 and papers therein or J. Alloys Comp. 490 (2010) 436). In addition, the photocatalytic degradation of methyl orange dye, under UV irradiation, is a benchmark reaction used to test the performance of any new nanosized TiO2 object (e.g., Appl. Surf. Sci., 257, 2011, 6451 or Thin Solid Films, 518, 2009, 16).

So, the manuscript is not suitable to be published in the current form, unless the Authors clearly indicate in which parts this work brings a new insight with respect to the state-of-the-art available so far.

Once this very major point is clarified, several other relevant parts need a careful revision.

1) Fig. 1. Did the Authors directly observe the formation of the intermediate nano-sheets? Otherwise, if they do not have a direct observation of these synthesis steps, some pictures can be considered rather speculative and should be avoided. Moreover, what does “scrolling” mean in this context?

2) Abstract and section 3.2. Performance values are reported as 80.00%. What is the error bar for this measurement? Are 4 significant digits necessary?

3) line 45. The Authors use the word “hierarchical”. Typically, this word indicates nanostructures where micro-, meso- and, sometimes, macro- pores or ordered patterns are present. However, most of the nano-objects mentioned on lines 47-48 are simply nanosized materials. The term is also repeated on lines 152 and 153: how can a material be “more hierarchical” than another one?

4) line 51-52. Ref. 22 mainly deals with TiO2 nanotubes synthesized via hydrothermal method, whereas anodized TiO2 tubes are rapidly cited in Table 1 of Liu’s work only. A deeper literature overview about successful syntheses of TiO2 nanotubes should be implemented. It can be useful for a proper comparison of the results of the present study with the previous literature.

5) line 59. The meaning of the word “adsorbent” is not fully clear. What kind of adsorbent were used as substrates?

6) Section 2.3. A better description of the experimental apparatuses and details is needed. For instance, which is the geometry of the photocatalytic reactor? Is the presence of the Ti foil, under the TiO2 layer, a problem for the diffusion of UV light?

7) Are the TiO2 materials hollow? Are they tubes? Or are they rods / whiskers? Actually, the presence of hollow tubes is not very clear in Fig. 8b and 8d. On line 168, it is stated that anatase TiO2 crystal phase is formed. However, classic tubes are not wide enough to display a clear crystalline pattern at XRD. A broader discussion on this point is necessary.

8) How uniform is the size of the tubes/rods? Can the Authors show a histogram with the distribution of diameters on a reasonably high number of tubes?

9) line 203. The improvement in the formation of TiO2 tubes is not likely related to the amount of energy provided during the process, but, rather, to the higher temperature, which makes the reaction faster.

10) Fig. 9b. The contact angle values in the line graph do not correspond to the ones described in the text (lines 253-254) and in the inset photographs.

11) Section 3.4. A comparison with a commercial P25 TiO2 material would help to understand the actual performance of these nanosized TiO2 obtained by hydrothermal method.

12) Fig. 10. What does “eta = 80%” mean? Is it the quantum yield of the photoreactor? If it is, how was it measured?

13) line 280. The Authors affirm that “a larger surface area” was provided, when the synthesis was performed at 160°C. Have they carried out a specific surface area analysis (e.g., by BET analysis)?

14) lines 281-284. The Authors affirm that the internal “particles” are not accessible. However, Fig. 7e shows a fine, but non-negligible series of openings. A HR-TEM micrograph of this material could help in understanding if the internal layers can be accessible or not by methyl orange molecules. However, as shown in Fig. 10b, at initial times (up to 30 min) the performance of the materials obtained at 160°C and 170°C are practically identical. Then, at longer times only, the material at 170°C reaches lower degradation values. Can it be attributed to some “fouling” of the surface by organic by-products, rather than to a diffusion limitation?

Minor points:

- line 219. “more stodgier” might be re-phrased as “denser”

-line 290 add “method” after “alkaline hydrothermal”

Author Response

Response to Reviewer 2 Comments

We highly appreciate reviewers’ valuable comments, constructive guidance and sincere contributions. We have carefully made specific responses point-by-point and revised the manuscript accordingly in light of the comments, which are annotated in RED words in the revised manuscript.

Reviewers' comments:

The manuscript describes the synthesis, characterization and one photocatalytic application of a series of TiO2 nanostructures grown over a metal Ti foil.The text is rather rich in experimental data. However, in its current form, it is not absolutely evident which elements of novelty are present with respect to the previous literature.

Actually, simple and highly reproducible approaches for the preparation of TiO2 nanotubes via hydrothermal method or via anodization of Ti foils have already been extensively described in the last decade (see, for instance, your Ref. 22 and papers therein or J. Alloys Comp. 490 (2010) 436). In addition, the photocatalytic degradation of methyl orange dye, under UV irradiation, is a benchmark reaction used to test the performance of any new nanosized TiO2 object (e.g., Appl. Surf. Sci., 257, 2011, 6451 or Thin Solid Films, 518, 2009, 16). So, the manuscript is not suitable to be published in the current form, unless the Authors clearly indicate in which parts this work brings a new insight with respect to the state-of-the-art available so far.

Response: This study used Ti foil as Ti source and subtract to develop TiO2 nanotubes network. The novelty of this work has been demonstrated as follows. 1. The TiO2 nanotubes network are grown directly on the Ti foil surface using alkaline hydrothermal method with excellent photocatalytic and hydrophilic properties. 2. The TiO2 nanotubes network and Ti foil are tightly connected, which facilitates recycling in the application to prevent secondary pollution. 3. The effect of NaOH concentration and hydrothermal temperature on TiO2 nanotubes network morphology and properties has been demonstrated.

Point 1:

Fig. 1. Did the Authors directly observe the formation of the intermediate nano-sheets? Otherwise, if they do not have a direct observation of these synthesis steps, some pictures can be considered rather speculative and should be avoided. Moreover, what does “scrolling” mean in this context?

Response 1: The formation of intermediate nanosheets can be directly observed, the specific formation process are shown in Figure 3. The “scrolling” mean is nanosheets

was washed with HCl and roll-up to form tubular structure.

Point 2:

Abstract and section 3.2. Performance values are reported as 80.00%. What is the error bar for this measurement? Are 4 significant digits necessary?

Response 2: Thank you for your reminding. The experiment of photocatalytic activity carried out 3 times, and the error bar was calculated and the results were added into the revised manuscript (line 29 and Figure 10).

Point 3:

line 45. The Authors use the word “hierarchical”. Typically, this word indicates nanostructures where micro-, meso- and, sometimes, macro- pores or ordered patterns are present. However, most of the nano-objects mentioned on lines 47-48 are simply nanosized materials. The term is also repeated on lines 152 and 153: how can a material be “more hierarchical” than another one?

Response 3: Thank you for your comment. The imprecise statement was corrected in the revised manuscript (line 46, 185).

Point 4:

line 51-52. Ref. 22 mainly deals with TiO2 nanotubes synthesized via hydrothermal method, whereas anodized TiO2 tubes are rapidly cited in Table 1 of Liu’s work only. A deeper literature overview about successful syntheses of TiO2 nanotubes should be implemented. It can be useful for a proper comparison of the results of the present study with the previous literature.

Response 4: Thank you for your reminding. The related References was corrected in the revised manuscript (Ref. 24). A deeper literature overview about successful syntheses of TiO2 nanotubes have been added into the revised manuscript (Line 51).

Point 5:

line 59. The meaning of the word “adsorbent” is not fully clear. What kind of adsorbent were used as substrates?

Response 5: Activated carbon and silica gel were used as substrates. The specific type of adsorbent have been added into the revised manuscript (line 62).

Point 6:

Section 2.3. A better description of the experimental apparatuses and details is needed. For instance, which is the geometry of the photocatalytic reactor? Is the presence of the Ti foil, under the TiO2 layer, a problem for the diffusion of UV light?

Response 6: The related content have been added into the revised manuscript (Section 2.3, Lin 101). The presence of the Ti foil under the TiO2 layer wasn't a problem for the diffusion of UV light, because the sample was placed flat in the quartz reaction tube and the UV source was directly above the quartz reaction tube.

Point 7:

Are the TiO2 materials hollow? Are they tubes? Or are they rods / whiskers? Actually, the presence of hollow tubes is not very clear in Fig. 8b and 8d. On line 168, it is stated that anatase TiO2 crystal phase is formed. However, classic tubes are not wide enough to display a clear crystalline pattern at XRD. A broader discussion on this point is necessary.

Response 7: The TEM image of TiO2 is relatively transparent, and the sample tends to be tubular structure. According to the XRD standard card of TiO2 (PDF NO. 78-2486), diffraction angles (2θ) 25.3° and 48.04° were separately characterized as the (101) and (200) planes for anatase TiO2. Other TiO2 crystal phase have different diffraction angles than anatase TiO2.

 Point 8:

How uniform is the size of the tubes/rods? Can the Authors show a histogram with the distribution of diameters on a reasonably high number of tubes?

Response 8: The related content have been added into the revised manuscript (Figure. 5f).

Point 9:

line 203. The improvement in the formation of TiO2 tubes is not likely related to the amount of energy provided during the process, but, rather, to the higher temperature, which makes the reaction faster.

Response 9: Thank you for your considerable review and comments. The statement was corrected in the revised manuscript (line 237).

Point 10:

Fig. 9b. The contact angle values in the line graph do not correspond to the ones described in the text (lines 253-254) and in the inset photographs.

Response 10: The contact angle values are consistent. You can check it. (Line 287 and Fig. 9b).

Point 11:

Section 3.4. A comparison with a commercial P25 TiO2 material would help to understand the actual performance of these nanosized TiO2 obtained by hydrothermal method.

Response 11: Thank you for your comments. The related content have been added into the revised manuscript (Line 330).

Point 12:

Fig. 10. What does “eta = 80%” mean? Is it the quantum yield of the photoreactor? If it is, how was it measured?

Response 12: The mean is the maximum degradation ratio of MO can reach 80.00 %.

Point 13:

line 280. The Authors affirm that “a larger surface area” was provided, when the synthesis was performed at 160°C. Have they carried out a specific surface area analysis (e.g., by BET analysis)?

Response 13: Thank you for your considerable review and comments. The number of TiO2 nanotubes were lower than that of 160°C when the temperature were 130°C -150(Fiure 7), TiO2 nanotubes maybe provide a larger surface area. Therefore, The imprecise statement was corrected in the revised manuscript (line 324).

Point 14:

lines 281-284. The Authors affirm that the internal “particles” are not accessible. However, Fig. 7e shows a fine, but non-negligible series of openings. A HR-TEM micrograph of this material could help in understanding if the internal layers can be accessible or not by methyl orange molecules. However, as shown in Fig. 10b, at initial times (up to 30 min) the performance of the materials obtained at 160°C and 170°C are practically identical. Then, at longer times only, the material at 170°C reaches lower degradation values. Can it be attributed to some “fouling” of the surface by organic by-products, rather than to a diffusion limitation?

Response 14: Thank you for your considerable review and comments. The presence of other by-products is less likely because the presence of other crystals is not observed in the XRD pattern of the sample, and only nanotubes are observed in the TEM image, and this is a relatively stable reaction process.

Reviewer 3 Report

The present paper describes the photocatalytic degradation of MO over TiO2 nanotubes. Unfortunately, experimental details of the procedure used for the photocatalytic experiments are nowhere to be found in the manuscript. I can therefore not judge their validity and must recommend to reject the manuscript.

Author Response

Response to Reviewer 3 Comments

We highly appreciate reviewers’ valuable comments, constructive guidance and sincere contributions. We have carefully made specific responses point-by-point and revised the manuscript accordingly in light of the comments, which are annotated in RED words in the revised manuscript.

Reviewers' comments:

The present paper describes the photocatalytic degradation of MO over TiO2 nanotubes. Unfortunately, experimental details of the procedure used for the photocatalytic experiments are nowhere to be found in the manuscript. I can therefore not judge their validity and must recommend to reject the manuscript.

Response: I am so sorry because my negligence has forgotten the part of the photocatalytic experiment written in the manuscript. After seeing your comment, the photocatalytic experiment part of TiO2 nanotubes have been added into the revised manuscript (line 121).

Round  2

Reviewer 1 Report

Some minor revisions only suggested to correct typos and mistakes in text and references. The properties and results are quite nice, so the manuscript is recommended for publication if other reviewers (and the editor) believe the manuscript is a good scope-match for the journal. The authors have considerably improved the manuscript, therefore I find it acceptable for publication.

Author Response

Response to Reviewer 1 Comments

Some minor revisions only suggested to correct typos and mistakes in text and references. The properties and results are quite nice, so the manuscript is recommended for publication if other reviewers (and the editor) believe the manuscript is a good scope-match for the journal. The authors have considerably improved the manuscript, therefore I find it acceptable for publication.

Response: Thank you for your approval.

Reviewer 2 Report

The Authors have addressed some of the doubtful points raised by the Reviewers. However, still some major flaws are present.

In particular, the points of novelty mentioned in the reply letter are only partial. The growth of TiO2 nanotubes on the Ti foil surface is not a first case and is also mentioned several times in some of the papers cited in the manuscript. Then, the proximity between TiO2 nanotubes and Ti foil is very common in nano-TiO2 obtained via anodization. Finally, the Authors affirm that they have studied the effect of NaOH concentration and temperature on TiO2 nanotubes morphology, but then the higher catalytic activity of the nanotubes obtained at 160°C is speculatively attributed to a larger surface area (“maybe”, line 319) without providing any experimental data in support to the hypothesis. In addition, none of these points were discussed nor added in the Introduction of the revised version of the manuscript. So, the original request about the actual “elements of novelty with respect to the previous literature” is still unsolved.

Moreover, a description of the geometry of the photoreactor, although requested at point 6, Reviewer 2, was not added. Such a description is fundamental to understand the efficacy and the reliability of the photocatalytic tests.

Similarly, no specific surface area analysis were carried out, although requested at point 13, Reviewer 2. Such information is crucial to carry out a proper comparison with P25 benchmark materials. Do P25 and nano-TiO2 (Fig. 3c) show remarkably different BET areas? If it is the case, it would be advisable to normalize the catalytic activity per unit of surface area. Only after a proper normalization, it would be possible to assess the higher or lower activity of the novel TiO2 systems.

As a minor point, for the numerical data, the standard deviation values have been added. However, are four significant digits necessary and reliable?

Thanking into account these major residual issues (especially, the statement of novelty), the revised manuscript cannot be accepted for publication in the present form.

Author Response

Response to Reviewer 2 Comments

We highly appreciate reviewers’ valuable comments, constructive guidance and sincere contributions. We have carefully made specific responses point-by-point and revised the manuscript accordingly in light of the comments, which are annotated in Yellow frame in the revised manuscript.

Reviewers' comments:

The Authors have addressed some of the doubtful points raised by the Reviewers. However, still some major flaws are present.

Point 1:

In particular, the points of novelty mentioned in the reply letter are only partial. The growth of TiO2 nanotubes on the Ti foil surface is not a first case and is also mentioned several times in some of the papers cited in the manuscript. Then, the proximity between TiO2 nanotubes and Ti foil is very common in nano-TiO2 obtained via anodization. Finally, the Authors affirm that they have studied the effect of NaOH concentration and temperature on TiO2 nanotubes morphology, but then the higher catalytic activity of the nanotubes obtained at 160°C is speculatively attributed to a larger surface area (“maybe”, line 319) without providing any experimental data in support to the hypothesis. In addition, none of these points were discussed nor added in the Introduction of the revised version of the manuscript. So, the original request about the actual “elements of novelty with respect to the previous literature” is still unsolved.

Response 1: Thank you for your considerable review and comments. BET of TiO2 nanotubes network are temporarily cannot carried out. The Chinese New Year is coming soon and school's testing centre have been closed for holiday. We can provide the experiment and results after Chinese New Yew. In this paper, The 500W high pressure mercury lamp was used as the light source. Other authors often use 100-300W high pressure mercury lamp as the light source in the selection of mercury lamp power. Moreover, the related literature have little research on the hydrophilic properties of TiO2 nanotubes prepared by alkaline hydrothermal method. In this paper, the detailed hydrophilic properties of TiO2 nanotubes prepared under different conditions were studied. The related points have been added into the revised Introduction (Line 73).

Point 2:

Moreover, a description of the geometry of the photoreactor, although requested at point 6, Reviewer 2, was not added. Such a description is fundamental to understand the efficacy and the reliability of the photocatalytic tests.

Response 2: Photochemistry reaction instrument (480mm × 420mm × 900mm) is a sealed rectangular parallelepiped structure. The related content have been added into the revised manuscript. (Line 117)

Point 3:

Similarly, no specific surface area analysis were carried out, although requested at point 13, Reviewer 2. Such information is crucial to carry out a proper comparison with P25 benchmark materials. Do P25 and nano-TiO2 (Fig. 3c) show remarkably different BET areas? If it is the case, it would be advisable to normalize the catalytic activity per unit of surface area. Only after a proper normalization, it would be possible to assess the higher or lower activity of the novel TiO2 systems.

Response 3: BET of TiO2 nanotubes network are temporarily cannot carried out. The Chinese New Year is coming soon and school's testing centre have been closed for holiday. We can provide the experiment and results after Chinese New Yew.

Point 4:

As a minor point, for the numerical data, the standard deviation values have been added. However, are four significant digits necessary and reliable?

Response 4: Thank you for your reminding. The four significant digits are data that are read directly on the instrument, so it is reliable and necessary. Other reviewers request that The degradation efficiencies should be discussed on the basis of degradation rate constants. Therefore, The related content was corrected in the revised manuscript (Figure. 10).

Reviewer 3 Report

The manuscript has been improved in the revision. However, the experimental details of the photocatalytic experiments still need to be clarified:

- The irradiation setup is insufficiently described. For independent validation and comparison, at least the irradiance (W/m² or photons/m²s) as well as the irradiated surface area and volume are needed. Even better would be to measure or calculate the incident volumetric photon flux density (in mol/Ls). Also, the spectrum of the employed lamp needs to be given.

- The comparison with literature data on the degradation of other dyes (3.4) is essentially meaningless as these were not conducted under the same equivalent photon flux. Please calculate (apparent) quantum efficiencies for proper comparison. Use initial degradation rates as the basis for these calculations.

- The degradation efficiencies should be discussed on the basis of degradation rate constants, obtained via kinetic analysis (first order or langmuir-hinshelwood), not %degradation, as the latter is highly non-linear.

- Equation 7 is incorrent and should be numbered 8.

Author Response

Response to Reviewer 3 Comments

We highly appreciate reviewers’ valuable comments, constructive guidance and sincere contributions. We have carefully made specific responses point-by-point and revised the manuscript accordingly in light of the comments, which are annotated in Yellow frame in the revised manuscript.

Reviewers' comments:

The manuscript has been improved in the revision. However, the experimental details of the photocatalytic experiments still need to be clarified:

Point 1:

The irradiation setup is insufficiently described. For independent validation and comparison, at least the irradiance (W/m² or photons/m²s) as well as the irradiated surface area and volume are needed. Even better would be to measure or calculate the incident volumetric photon flux density (in mol/Ls). Also, the spectrum of the employed lamp needs to be given.

Response 1: The irradiation intensity was not detected during the photocatalytic degradation experiment. The Chinese New Year is coming soon and school have been closed for holiday. We can provide the experiment and results after Chinese New Yew. The irradiated surface area were 0.00956 m2. The employed lamp with a main wavelength of 365 nm(Line 129).

Point 2:

The comparison with literature data on the degradation of other dyes (3.4) is essentially meaningless as these were not conducted under the same equivalent photon flux. Please calculate (apparent) quantum efficiencies for proper comparison. Use initial degradation rates as the basis for these calculations.

Response 2: Thank you for your comment. Some authors' data cannot be found in their articles, so it is impossible to calculate (apparent) quantum efficiency, and remove the related content of degradation organic dyes.

Point 3:

The degradation efficiencies should be discussed on the basis of degradation rate constants, obtained via kinetic analysis (first order or langmuir-hinshelwood), not %degradation, as the latter is highly non-linear.

Response 3: Thank you for your reminding. The imprecise statement was corrected in the revised manuscript (Figure. 10).

Point 4:

Equation 7 is incorrent and should be numbered 8.

Response 4: Thank you for your review and comments. The imprecise numbered was corrected in the revised manuscript (Line 164).

Round  3

Reviewer 2 Report

Reviewers 2 and 3 have stressed the importance of BET measurements irradiation intensity tests. These additional tests can be carried out after the Chinese holiday time, as suggested by the Authors. We look forward to having the final manuscript with the required data soon.

However, I do not agree on point 4 for Reviewer 2. The output of a 4-digit instrument does not mean that the accuracy of the measurement is of 4 digits as well. The Authors should estimate the error bar of any measurement and use the number of significant digits accordingly.

Reviewer 3 Report

Response 1: The irradiation intensity was not detected during the photocatalytic degradation experiment. The Chinese New Year is coming soon and school have been closed for holiday. We can provide the experiment and results after Chinese New Yew. The irradiated surface area were 0.00956 m2. The employed lamp with a main wavelength of 365 nm(Line 129).

Then please resubmit the manuscript after you have conducted the necessary experiments.